# The vortex-driven dynamics of droplets within droplets

A. Tiribocchi [1,2 ✉], A. Montessori[2], M. Lauricella[2], F. Bonaccorso[1,2], S. Succi[1,2,3], S. Aime[3,4], M. Milani[5] &
D. A. Weitz[3,6]

Understanding the fluid-structure interaction is crucial for an optimal design and manu-facturing of soft mesoscale materials. Multi-core emulsions are a class of soft fluids assembled from cluster configurations of deformable oil-water double droplets (cores), often employed as building-blocks for the realisation of devices of interest in bio-technology, such as drug-delivery, tissue engineering and regenerative medicine. Here, we study the physics of multi-core emulsions flowing in microfluidic channels and report numerical evidence of a surprisingly rich variety of driven non-equilibrium states (NES), whose formation is caused by a dipolar fluid vortex triggered by the sheared structure of the flow carrier within the microchannel. The observed dynamic regimes range from long-lived NES at low core-area fraction, characterised by a planetary-like motion of the internal drops, to short-lived ones at high core-area fraction, in which a pre-chaotic motion results from multi-body collisions of inner drops, as combined with self-consistent hydrodynamic interactions. The onset of pre-chaotic behavior is marked by transitions of the cores from one vortex to another, a process that we interpret as manifestations of the system to maximize its entropy by filling voids, as they arise dynamically within the capsule.

[1] Center for Life Nano Science@La Sapienza, Istituto Italiano di Tecnologia, Roma 00161, Italy. [2] Istituto per le Applicazioni del Calcolo CNR, via dei Taurini 19, Rome 00185, Italy. [3] Institute for Applied Computational Science, John A. Paulson School of Engineering and Applied Sciences, Harvard University, Cambridge, MA 02138, USA. [4] Matiére Molle et Chimie, Ecole Supérieure de Physique et Chimie Industrielles, Paris 75005, France. [5] Universitá degli Studi di Milano, via Celoria 16, Milano 20133, Italy. [6] Department of Physics, Harvard University, Cambridge, MA 02138, USA. ✉email: adriano.tiribocchi@iit.it

Recent advances in microfluidics have highlighted the possibility to design highly ordered, multi-core emulsions in an unprecedentedly controlled manner[1–12]. These emulsions are hierarchical soft fluids, consisting of small drops (often termed "cores") immersed within larger ones, and stabilized over extended periods of time by a surfactant confined within their interface.

A typical example is a collection of immiscible water droplets, embedded within a surrounding oil phase[13]. This is usually manufactured in a two-step process, by first emulsifying different acqueous solutions in the oil phase and then encapsulating water drops within the same oil phase in a second emulsification step[1,9,10].

Due to their peculiar tiered architecture, they have served as templates to manufacture microcapsules with a core–shell geometry that have found applications in a number of sectors of modern industry, such as in food science for the realisation of low calory dietary products and encapsulation of flavours[14–17], in pharmaceutics for the delivery and controlled release of substances[18–22], in cosmetics for the production of personal care items[6,23–25] and in tissue engineering, as building-blocks for the design of tissue-like soft porous materials[26–28]. More recently, they have also been used as a tool to mimic cell–cell interactions within a dynamic environment provided by flowing capsules in microcapillary channels[29,30].

Understanding their behavior in the presence of fluid flows, even under controlled experimental conditions, remains a crucial requirement for a purposeful design of such functionalized materials. The rate of release of the drug carried by the cores, for example, is significantly influenced by surfactant concentration and hydrodynamic interactions[31]. The latter ones, even when moderate, can foster drop collisions as well as shape deformations that may ultimately compromise the release and the delivery towards targeted diseased tissues. Controlling mechanical properties as well as long-range hydrodynamic interactions of the liquid film formed among cores is of paramount importance for ensuring the prolonged stability of food-grade multiple emulsions[32]. This is essential in high internal phase emulsions extensively used in tissue engineering, where such forces can considerably alter pore size and rate of polydispersity[28,33], thus jeopardizing the structural homogeneity of the material.

Building ad hoc mathematical and computational models is therefore fundamental to make progress along this direction. Indeed, in stark contrast with the impressive advances in the experimental realisation of multi-core emulsions[9,10,34,35], it is only recently that efforts have been directed to the theoretical investigation of the rheology resulting from the highly non-linear fluid–structure interactions taking place in such systems[36–38].

Continuum theories, combined with suitable numerical approaches (such as lattice Boltzmann methods[39–41] and boundary integral method[38]) have proven capable to capture characteristic features observed in double emulsion experiments, such as their production within microchannels[40], the typical shape deformations of the capsule (elliptical and bullet-like) under moderate shear flows[42–44], as well as more complex dynamic behaviors, such as the breakup of the enveloping shell occurring under intense fluid flows[45]. However, much less is known about the dynamics of more sophisticated systems, such as multiple emulsions with distinct inner cores, theoretically investigated only by a few authors to date[38,39].

A befitting theoretical framework for describing their physics can be built on well-established continuum principles, whose details are illustrated in the "Methods" section. It is essentially based on a phase field approach[39,46–48], in which a set of passive scalar fields $\phi_i(\mathbf{r}, t)$ ($i = 1, …, N_d$, where $N_d = N + 1$ is the total number of droplets and $N$ is the number of cores) accounts for the density of each droplet, while a vector field $\mathbf{v}(\mathbf{r}, t)$ represents the global fluid velocity. The dynamics of each field $\phi_i$ is governed by a set of Cahn–Hilliard equations, while the velocity obeys the Navier–Stokes equations[49,50]. The thermodynamics of the mixture is encoded in a Landau free-energy functional augmented with a term preventing the coalescence of the cores, thus capturing the repulsive effect produced by a surfactant adsorbed onto a droplet interface.

In this paper, we employ the aforementioned method to numerically study the pressure-driven flow of multi-core droplets confined in a microchannel, following a design directly inspired to actual microfluidic devices. Extensive lattice Boltzmann simulations provide evidence of a rich variety of driven nonequilibrium states (NES), from long-lived ones at low droplet area fraction (and low number of inner drops), characterised by a highly correlated, planetary-like motion of the cores, to short-lived ones at high droplet area fraction (with moderate/high number of cores), in which multiple collisions and intense fluid flows trigger a chaotic-like dynamics. Central to each of these non-equilibrium states is the onset of a vorticity dipole within the capsule, which arises as an inevitable consequence of the sheared structure of the velocity field within the microchannel. Such dipole structure naturally invites a classification of these states in close analogy with the statistical mechanics of occupation numbers. Even though our system is completely classical, such representation discloses a transparent and insightful interpretation of the transition from periodic to quasi-chaotic steady-states, in terms of level crossings between the occupation numbers in the two vortex structures. Such level crossings are interpreted as manifestations of the system to maximise its entropy by filling voids, which arise dynamically within the multibody structure resulting from the self-consistent motion of the cores within the capsule. This is consistent with the notion of entropy as propensity to motion rather than microscopic disorder[51].

## Results

**Fluid–structure interaction in a core-free emulsion.** We start by describing droplet shape and pattern of the fluid velocity at the steady state in a core-free emulsion under a Poiseuille flow. Once this is imposed, the droplet, initially at equilibrium (Fig. 1a), acquires motion, driven by a constant pressure gradient $\Delta p$ applied across the longitudinal direction of the microfluidic channel. The resulting fluid flow as well as the shape of the emulsion are controlled by the capillary and the Reynolds numbers, defined as Ca = $\frac{v_{max}\eta}{\sigma}$ and Re = $\frac{\rho v_{max} D_O}{\eta}$. Here $v_{max}$ is the maximum value of the droplet speed, $\eta$ is the shear viscosity of the fluid, $\sigma$ is the surface tension, $\rho$ is the fluid density, and $D_O$ is the diameter of the shell

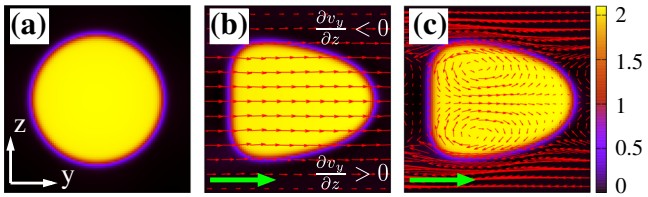

**Fig. 1 Steady-state shapes and velocity field structure of a core-free emulsion under Poiseuille flow. a** Equilibrium configuration of a core-free emulsion. **b** and **c** Steady-state shapes at Re $\simeq$ 3 and Ca $\simeq$ 0.8. Red arrows indicate the direction of the fluid flow computed in the lab frame (**b**) and with respect to the center of mass of the droplet (**c**). Here two eddies rotating clockwise (bottom) and counterclockwise (top) emerge within the droplet. The droplet radius at equilibrium is $R = 30$ and the color map represents the value of the order parameter $\phi$, ranging between 0 (black) and 2 (yellow). This applies to all figures in the paper.

(taken as characteristic length of the multi-core emulsion). In our simulations Ca roughly varies between 0.02 and 1 and Re may range from 0.02 and 5, hence inertial effects are mild and the laminar regime is generally preserved.

In Fig. 1b and c we show an example of shape of a core-free emulsion at the steady state (see Supplementary Movie 1 for the full dynamics). In agreement with previous studies[52–56], the droplet attaines a bullet-like form, more stretched along the flow direction for higher values of Re and Ca (i.e. larger pressure gradients). Such shape results from the parabolic structure of the flow profile (see Supplementary Figs. 1 and 2 for further details about the steady-state velocity profile), moving faster in the center of the channel and progressively slower towards the wall (Fig. 1b). If computed with respect to the droplet frame, the flow field exhibits two symmetric counterrotating eddies, resulting from the confining interface of the capsule and whose direction of rotation is consistent with a droplet moving forward (rightwards here, see Fig. 1c). These structures are due to the gradient of $v_y$ along the $z$ direction, a quantity positive within the lower half of the emulsion and negative in the upper. As long as the droplet remains core-free, such fluid recirculations (also observed, for instance, in micro-emulsions propelled either through Marangoni effect[57–59] or by means of an active material, such as actomyosin proteins[60–62], dispersed within) are stable, and their pattern remains basically unaltered.

One may wonder whether this picture still holds when small cores are encapsulated. This essentially means understanding (i) to what extent the dipolar fluid flow structure is stable when the effective area fraction of the internal droplets $A_c = \frac{N\pi R_i^2}{\pi R_O^2}$ (where $R_i$ and $R_O$ are the radii of the cores and of the shell) increases, and (ii) how the coupling between the fluid velocity $\mathbf{v}$ and a number of passive scalar fields $\phi_i$ affects the dynamics of the multi-core emulsion.

In the next sections we show that the double eddy fluid structure is substantially preserved, although modifications to this pattern occur when $A_c$ is larger than roughly 0.35. More specifically, while in a double emulsion ($N = 1$) the stream in the middle of the droplet drives the core at the front-end of the shell where the core gets stuck, when $N > 1$ the vortices trigger and sustain a persistent periodic motion of the cores within each half of the emulsion. As long as $A_c < 0.35$ (achieved with $N = 3$),

cores remain confined within either the upper or the lower part of the emulsion giving rise to long-lived nonequilibrium steady states. When $A_c > 0.35$ ($N \geq 4$) droplet crossings between the two regions may occur, and short-lived aggregates of three or more cores only temporarily survive. Hence, the whole emulsion can be effectively visualized as a two-state system in which the two eddies foster the motion of the cores and crucially affect the duration of the states.

By using the tools of statistical mechanics, we propose a classification of such states in terms of the occupation number formalism, where $\langle \alpha_1, \ldots, \alpha_j | \alpha_{j+1}, \ldots, \alpha_N \rangle$ represents a state in which $j$ and $N-j$ distinct cores occupy, respectively, the upper and the lower half of the emulsion, with $j = 1,.., N$. This is analogous to determine the number of ways $N$ distinguishable particles can be placed in two boxes.

**Classification of states.** In Fig. 2a–f, we show the equilibrium configurations of a single core (Fig. 2a, $N = 1$), a two-core (Fig. 2b, $N = 2$), a three-core (Fig. 2c, $N = 3$), a four-core (Fig. 2d, $N = 4$), a five-core (Fig. 2e, $N = 5$), and a six-core (Fig. 2f, $N = 6$) emulsion. Once a Poiseuille flow is applied, the emulsions attain a steady state in which, unlike the core-free droplet of Fig. 1, the dynamics of the cores and the velocity field are crucially influenced by the effective area fraction $A_c$ and the number of cores $N$. In Fig. 3 we show a selection of the NES observed.

**Long-lived non-equilibrium states.** The simplest configuration is the one in which $N = 1$. In this case the core and the shell are initially advected forward (rightwards in the figure) and, at the steady state, the former gets stuck at the front-end of the latter (see, for example, Supplementary Movie 2). In Fig. 3, two examples for slightly different values of Re and Ca are shown. Note incidentally that, in agreement with previous studies[39,40,42,44,52,53,63], as long as Re $\simeq 1$, the internal core, unlike the interface of the external shell, is only mildly affected by the fluid flow. This is due to its higher surface tension induced by the smaller curvature radius, which prevents relevant shape deformations.

This picture is dramatically altered when the number of cores increases. If $N = 2$ ($A_c \sim 0.18$), for example, two nonequilibrium long-lived states emerge. In the first one (Fig. 3c and Supplementary

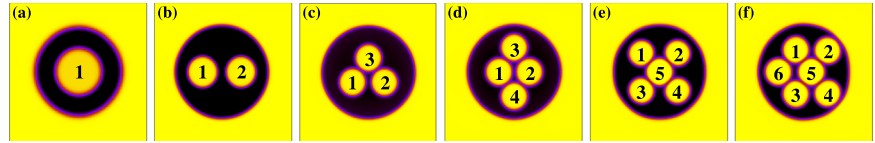

**Fig. 2 Equilibrium states of multi-core emulsions.** Equilibrium configurations of a single-core (**a**), a two-core (**b**), a three-core (**c**), a four-core (**d**), a five-core (**e**), and a six-core (**f**) emulsion. Droplet radii are as follows: $R_i = 15$, $R_O = 30$ (**a**), $R_i = 17$, $R_O = 56$ (**b–f**).

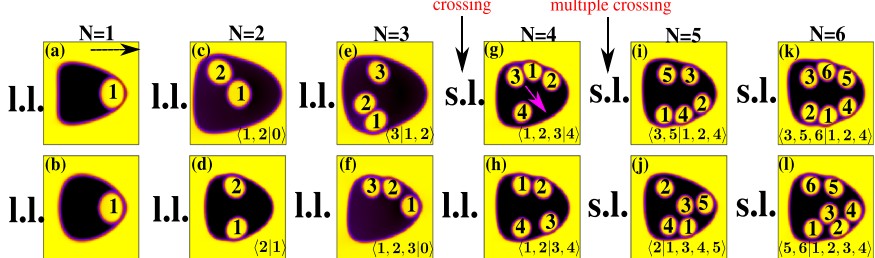

**Fig. 3 Nonequilibrium states of multi-core emulsions under Poiseuille flow.** The dotted black arrow indicates the flow direction (which applies to all cases) while $N$ represents the number of cores encapsulated. **a** Is obtained for Re $\simeq 3$ and Ca $\simeq 0.85$, while **b** for Re $\simeq 1.2$ and Ca $\simeq 0.35$. All the other cases correspond to Re $\simeq 3$ and Ca $\simeq 0.85$. As long as $A_c < 0.35$, the nonequilibrium states are long-lived (l.l.), while if $A_c > 0.35$ they turn into short-lived (s.l.). Here, crossings of cores from one region towards the other one start to occur. The magenta arrow indicates the direction of a crossing. Multiple crossings are observed as $A_c$ increases. In each snapshot, states are indicated by means of the occupation number classification.

Movie 3), the two cores remain locked in the upper (or lower) part of the emulsion, where the fluid eddy fosters a periodic motion in which each core chases the other one in a coupled-dance fashion (see the next section for a detailed description of this dynamics). In the second one (Fig. 3d and Supplementary Movie 4), the two cores remain separately confined within the top and the bottom of the emulsion, and move, weakly, along circular trajectories. By using the statistics of occupation number, we indicate with $\langle 1, 2|0\rangle$ the state in which cores 1 and 2 are in the upper region while the lower one is empty, and with $\langle 2|1\rangle$ the state where cores 2 and 1 occupy, separately, each half.

More complex effects emerge when $A_c$ is further increased. If $N = 3$ ($A_c \sim 0.27$), once again we find two different long-lived NES, namely $\langle 3|1, 2\rangle$ and $\langle 1, 2, 3|0\rangle$. In the first one (Fig. 3e and Supplementary Movie 5), the core 3 is confined at the top of the emulsion and moves following a circular path, while cores 1 and 2 remain at the bottom of the emulsion and reproduce the coupled-dance dynamics observed for $N = 2$. In the second one (Fig. 3f and Supplementary Movie 6), the three cores, sequestered in the upper region, exhibit a more complex three-body periodic motion, whose dynamics is discussed later. However, although long-lasting, such state may turn unstable due to hydrodynamic interactions and to direct collisions with other cores. This is precisely what happens when $N$ and $A_c$ are further augmented.

**Short-lived NES.** Indeed, when $N = 4$ ($A_c \sim 0.37$), we find a state in which three cores move in one region and a single core within the other one. This is indicated as $\langle 1, 2, 3|4\rangle$ (Fig. 3g). However, this state lives for a short period of time since droplet 3 crosses from the top towards the bottom of the emulsion and produces the long-lived nonequilibrium state $\langle 1, 2|3, 4\rangle$, in which couples of cores ceaselessly dance within two separate regions (Fig. 3e and Supplementary movie 7). Such transition, indicated as $\langle 1, 2, 3|4\rangle \rightarrow \langle 1, 2|3, 4\rangle$, occurs due to the high values of $A_c$, generally larger than 0.35. This means that, within half of the emulsion, the effective area fraction is even higher (more than 0.5), and the three-core state would be unstable to changes of the flow direction and to unavoidable collisions with the other cores. This is the reason why, for example, the state $\langle 1, 2, 3, 4|0\rangle$, although realizable in principle, has not been observed.

For higher values of $N$, $A_c$ further increases and multiple crossings occur. If $N = 5$ ($A_c \sim 0.46$) for instance, short-lived dynamical states appear, such as $\langle 3, 5|1, 2, 4\rangle$ or $\langle 2|1, 3, 4, 5\rangle$, in which four cores are temporarily packed within half of the emulsion (Fig. 3i, j and Supplementary Movie 8). However, these states are highly unstable since cores cross the two regions of the emulsion multiple times. Finally, more complex short-lived states are observed when $N = 6$ ($A_c \sim 0.55$), such as $\langle 3, 5, 6|1, 2, 4\rangle$ and $\langle 5, 6|1, 2, 3, 4\rangle$ (Fig. 3k, l and Supplementary Movie 9).

In the next section we discuss more specifically the dynamics of these states focussing, in particular, on how the fluid-structure interaction affects the droplets motion.

**Two-core emulsion.** We begin from the two-core emulsion, in which two droplets are initially encapsulated as in Fig. 2b and attain the state $\langle 1, 2|0\rangle$. Like the single-core case, once the flow is applied both cores and external droplet are dragged forward by the fluid. However, while the latter rapidly attains its steady state (see Fig. 4a in which an instantaneous configuration is shown for Re $\simeq 3$, Ca $\simeq 0.52$), the internal core placed at the rear side

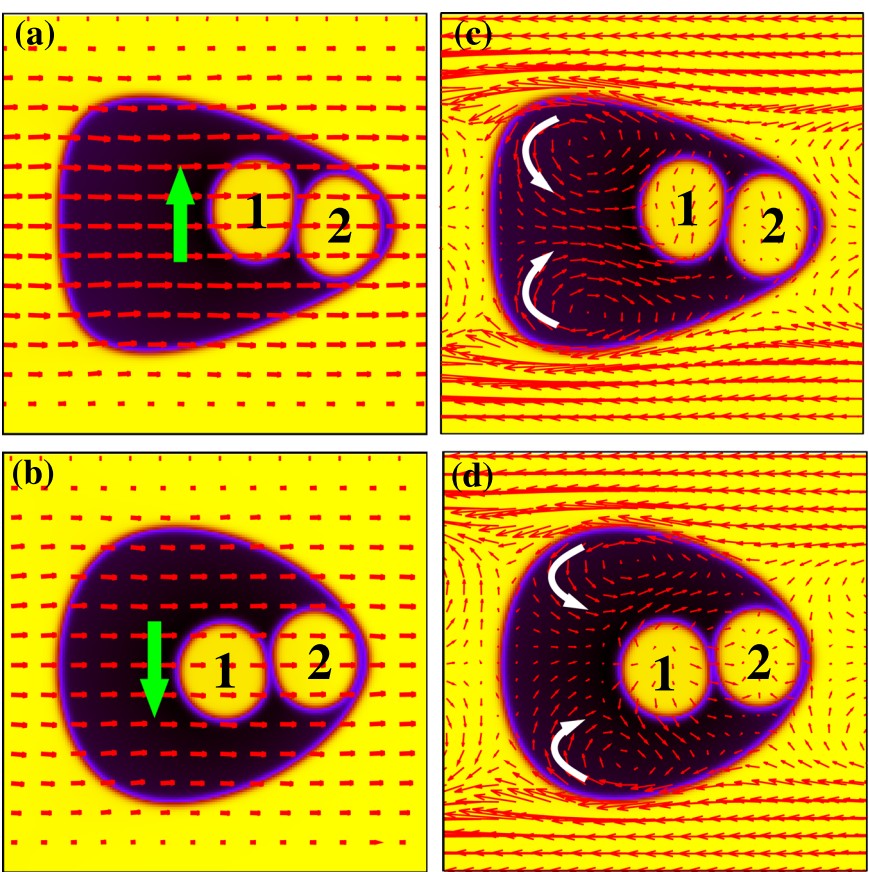

**Fig. 4 Onset of periodic motion in a two-core emulsion.** In **a–c** Re $\simeq 3$ and Ca $\simeq 0.52$, while in **b–d** Re $\simeq 1.2$ and Ca $\simeq 0.2$. Red arrows indicate the direction of the fluid flow in the lab frame (**a** and **b**), and in the external droplet frame (**c** and **d**). Green arrows in **a** and **b** denote the direction, perpendicular to the flow, along which the droplet 1 starts its motion, while white arrows in **c** and **d** bespeak the direction of the fluid recirculations.

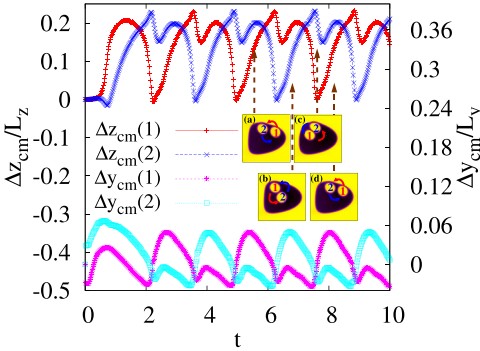

**Fig. 5 Time evolution of the displacement $\Delta z_{cm}$ and $\Delta y_{cm}$ of the cores.** On the left axis $\Delta z_{cm}(i) = z_{cm}(i) - z_{cm}(O)$ (red/plusses and blue/crosses) and on the right axis $\Delta y_{cm}(i) = y_{cm}(i) - y_{cm}(O)$ (pink/asterisk and cyan/squares) of the two cores. They are calculated with respect to the center of mass of the external droplet. The inset shows the typical trajectories of the cores during a cycle. Red and blue arrows indicate the direction of rotation of droplets 1 and 2, respectively. Snapshots are taken at $t = 5.54 \times 10^5$ (**a**), $t = 6.3 \times 10^5$ (**b**), $t = 7.5 \times 10^5$ (**c**), and $t = 8.16 \times 10^5$ (**d**). Multiplication by a factor $10^5$ is understood for the simulation time $t$ on the horizontal axis. This applies to all plots in the paper.

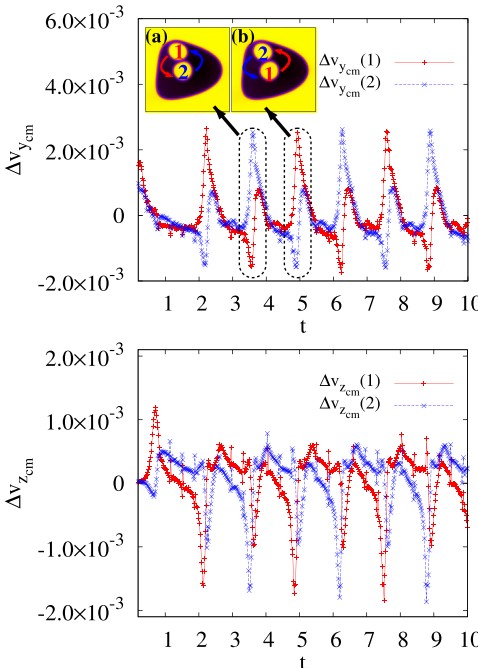

**Fig. 6 Speed of the cores.** Time evolution of the $y$ and $z$ components of the velocity of center of mass of the two cores computed with respect to that of the external droplet. The inset highlights the position of the inner drops at the points of inversion of motion. Red and blue arrows indicate their direction of motion. Snaphots are taken at $t = 3.6 \times 10^5$ (**a**) and $t = 4.94 \times 10^5$ (**b**) timesteps.

(core 1) approaches the one at its front (core 2), since it moves slightly faster due to the larger flow field in the middle line of the emulsion (see Fig. 4, and position and speed of their center of mass for $t < 10^5$ in Figs. 5 and 6). This results in a dynamic chain-like aggregate of droplets, moving together towards the leading edge of the external interface. Note that droplet merging is inhibited, since the repulsion among cores (mimicking the effect of a surfactant) is included (see section "Methods").

However, such aggregate is unstable to weak perturbations of the flow field, an effect due to the non-trivial coupling between

the velocity of the fluid and the interfaces of the cores, and essentially governed by a term of the form $\nabla \cdot (\phi_i \mathbf{v})$ (see Eq. (2) in section "Methods"). In Fig. 4 we show the typical flow field in the lab frame (Fig. 4a and b) and in the external droplet frame (Fig. 4c and d) at the onset of this instability for two different values of Re and Ca. Where the former looks approximately laminar, the latter, once again, exhibits two separate counter-rotating patterns, in the upper and in the lower part of the emulsion. Unlike the core-free emulsion, here the flow field in the middle departs from the typical homogeneous and unidirectional structure, especially within and in the surroundings of the cores where it fosters the motion of the drops either upwards (Fig. 4a–c) or downwards (Fig. 4b–d).

Once this occurs, the fluid vortices capture the cores and sustain a persistent and periodic circular motion of both of them around a common center of mass, confined within the lower or the upper region of the emulsion. Such dynamics is described in Fig. 5, where we show the time evolution of the displacement $\Delta z_{cm}(i) = z_{cm}(i) - z_{cm}(O)$ and $\Delta y_{cm}(i) = y_{cm}(i) - y_{cm}(O)$ of the cores with respect to the center of mass of the external droplet, at Re $\simeq 3$ and Ca $\simeq 0.52$.

After being driven forward in the middle of the emulsion, the cores acquire motion upwards and backwards near the external interface, with the droplet at the rear (1) closely preceding the front one (2). Subsequently, the circular flow pushes both cores back towards the center of the emulsion, but while the former (1) moves towards the leading edge, the latter (2) follows a shorter circular counterclockwise trajectory which allows to overtake the other core. The process self-repeats periodically, alternating, at each cycle, the core leading the motion.

In Fig. 6 we show the $y$ and $z$ components of the center of mass velocity (in the external droplet frame) of both cores. They have a cyclic behavior with the same periodicity, although $\Delta v_{z_{cm}} = v_{z_{cm}}(i) - v_{z_{cm}}(O)$ shows an asymmetric pattern. This occurs because, when moving upwards (during the first half of the cycle), the core has a lower velocity (approximately $5 \times 10^{-4}$ in simulation units) than when moving back towards the bulk (roughly—$2 \times 10^{-3}$). This is not the case of $\Delta v_{y_{cm}} = v_{z_{cm}}(i) - v_{z_{cm}}(O)$, which, on the contrary, exhibits a symmetric structure, with maxima and minima attained at the center of the emulsion and at the top, respectively.

Finally, note that the initial location of the cores importantly affects the dynamic response of the emulsion. Indeed, if cores 1 and 2 are originally aligned vertically, rather than horizontally, the state $\langle 2|1 \rangle$ is obtained. Once the flow is applied, they acquire a circular motion triggered by the eddies but, since they are placed within two separate sectors (top and bottom of the emulsion) from the beginning, they interact only occasionally along the middle region of the emulsion.

**Three-core emulsion**. We first discuss the dynamics of the state $\langle 3|1, 2 \rangle$. In Fig. 7 we show the time evolution of the displacement of three cores, initially placed as in Fig. 2, under a Poiseuille flow when Re $\simeq 3$ and Ca $\simeq 0.52$. Here cores 1 and 2 are captured by the eddy formed downward and, like the state $\langle 1, 2|0 \rangle$, they rotate periodically around approximately circular orbits. On the other hand, core 3 is locked upward, where it rotates along a shorter rounded trajectory at a higher frequency, roughly twice larger than that of cores 1 and 2, but basically at the same speed ($\mathcal{O}(10^{-3})$ in simulation units). This is shown in Fig. 8a–c, where the time evolution of the speed of each core (computed with respect to that of the external droplet) is reported. Although, at the steady state, two fluid eddies can be clearly distinguished (see Fig. 8d), the pattern of the lower one deviates from the typical rounded shape due to the presence of the cores. In the next section we will show that such distortions will consolidate when

$A_c$ (and $N$) increases, and will essentially favour crossings of cores between the two regions of the emulsion.

A dynamics in which two cores rotate in the upper region and the other core in the lower one (such as the state $\langle 1, 2|3\rangle$) can be observed, for instance, for smaller values of Re and Ca, and shares essentially the same features with the previous case.

Finally, if the three cores are initially confined within the upper half of the emulsion, al late times the state $\langle 1, 2, 3|0\rangle$ is attained (see Fig. 3f and Supplementary Movie 6). Once again, the internal cores exhibit a periodic motion along roughly circular trajectories, although the reciprocal interactions complicate the dynamics. Shortly, the core at the rear, 1, is initially pushed forward by the flow in the middle of the emulsion, while cores 2 and 3 get locked upwards and rotate synchronized (like the state $\langle 1, 2|0\rangle$). Afterwards, core 1 progressively shifts towards the upper region, connects with the other two to form a three-core chain moving coherently. Such aggregate is only partially broken when a core approaches the middle of the emulsion (where an intense flow current pushes it forward), but is rapidly reshaped when the core, once again, migrates back upwards.

Such dynamic behavior suggests, once more, that the initial position of the cores plays a crucial role in driving the dynamics of the emulsion towards a targeted steady state. Indeed, the formation of the state $\langle 1, 2, 3|0\rangle$, starting, for example, from Fig. 2c, would require crossings of cores from the bottom towards the top of the emulsion, a very unlikely event if $A_c$ remains low.

However, this state may turn to unstable and decay into a long-lived one when $A_c$ augments. This is what happens if four cores are included.

**Four cores: Droplet crossings and short-lived clusters.** In Fig. 9a we show the time evolution of the displacement $\Delta z_{cm}$ of four cores originally encapsulated as in Fig. 2d. While droplets 1 and 2, initially carried rightwards by the fluid, are then driven towards the upper part of the emulsion where soon acquire the usual dance-like periodic motion, droplet 3 travels towards the bottom driven by an intense, mainly longitudinal, flow (Fig. 9b, c), which significantly alters the typical double eddy pattern of the velocity fluid (see Supplementary Fig. 3 for a detailed structure of the velocity field).

After the transition a clear fluid recirculation is definitively restored and favours, once again, the onset of a persistent coupled motion occurring in a similar manner as the others: each droplet chases the other one and, recursively, the droplet at the front (say 4) is pushed backwards (i.e. towards the leading edge of the emulsion) along a circular trajectory larger than the one covered by the droplet at the rear (say 3), which, now, leads to motion (see Supplementary Movie 7).

At the steady-state, the four cores exhibit a long-lived coupled dance in pairs of two, occurring without further crossings and within two separate regions of the emulsion. These results suggest that, although for a short period of time, the periodicity of the motion can be temporarily lost when $A_c$ is sufficiently high.

This steady-state dynamics looks rather robust, since, unlike the three-core emulsion, occurs even if the cores are initially confined, for example, within the lower half of the emulsion (see Supplementary Movie 8, where cores are numbered the same as in Fig. 2d). Despite the asymmetric starting configuration, the core on top of the others (i.e. 1) and the one on the back (i.e. 3) are quickly driven towards the leading edge of the emulsion by the flow in the middle and are then captured by the fluid vortex in un upper region. Thus here two cores, rather than one, actually exhibit a crossing, although this occurs soon after the Poiseuille flow sweeps over the droplet. A steady state of the form $\langle 1, 3|2, 4\rangle$ emerges and lasts for long periods of time.

In the next section we show that when $A_c$ attains a value equal to (or higher than) 0.4, the dynamics of the cores lacks of any periodic regularity and the motion turns to chaotic.

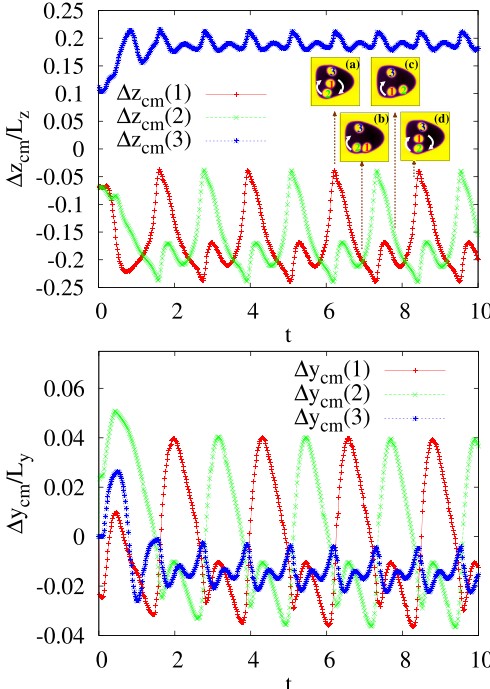

**Fig. 7 Time evolution of $\Delta z_{cm}$ and $\Delta y_{cm}$ in a three-core emulsion.** Cores 1 and 2 show a periodic motion around a common center of mass in the bottom region of the emulsion while core 3 travels persistently along a circular path confined in the upper part. The inset shows four instantaneous configurations of $\phi_i$ during a periodic cycle. White arrows denote the trajectories of the cores. Snapshots are taken at $t = 6.22 \times 10^5$ (**a**), $t = 6.96 \times 10^5$ (**b**), $t = 7.8 \times 10^5$ (**c**), $t = 8.42 \times 10^5$ (**d**).

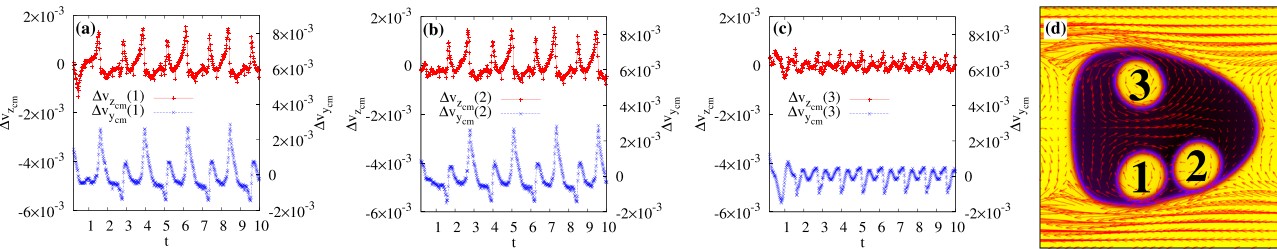

**Fig. 8 Speed of the cores and fluid velocity structure of the emulsion. a**–c Time evolution of $\Delta v_{z_{cm}}$ (red/plusses) and $\Delta v_{y_{cm}}$ (blue/crosses) of the cores, computed with respect to the speed of center of mass of the external droplet. **d** Typical pattern of the velocity field calculated with respect to the center of mass velocity of the external droplet. Two large eddies dominate the dynamics although relevant distortions, due to the presence of the cores, are mainly produced in the lower part of the emulsion.

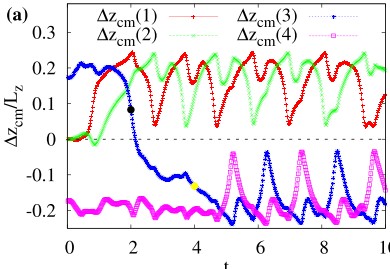 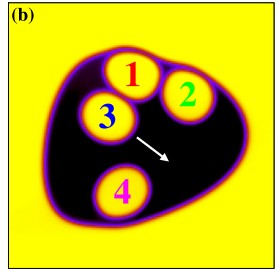 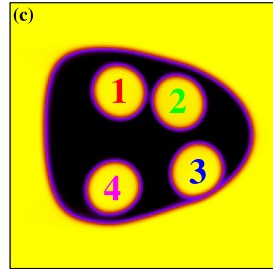

**Fig. 9 Level crossing in a four-core emulsion. a** Time evolution of the displacement $\Delta z_{cm}$ of four cores. Drops 1 and 2 (red/plusses and green/crosses, respectively) exhibit a periodic motion in the upper part of the emulsion whereas cores 3 and 4 (blue/asterisk and pink/square) in the lower one, once a crossing has occurred. The black dotted line is a guide for the eye representing the separation between the two regions of the emulsion. Panels **b** and **c** show the positions of the internal droplets during the crossing of droplet 3. They are taken at $t = 2 \times 10^5$ (**a**) and $t = 4 \times 10^5$ (**b**), marked by two spots (black and yellow) in (**a**). A white arrow indicates the direction of motion of droplet 3.

**Five and six cores: Onset of non-periodic dynamics**. In Fig. 10 we show the time evolution of the displacement $\Delta z_{cm}$ in a five (Fig. 10a–e) and six-core (Fig. 10f–k) emulsion, where $A_c \simeq 0.46$ and $A_c \simeq 0.55$, respectively (see also Supplementary Movies 9 and 10).

In such systems, only short-lived states are observed (such as those shown in Fig. 3i–l), since multiple crossings occur between the top and the bottom of the emulsion. An enduring dance-like dynamics, for instance, is observed only for pair of cores and lasts for relatively shorts times, interrupted by crossings taking place within the emulsion. When this event occurs, the incoming droplet temporarily binds with the others, yielding to a short-lived nonequilibrium steady state such as those shown in Fig. 3, in turn destroyed as soon as a further core approaches. Importantly, such complex dynamics almost completely removes any periodicity of the motion of the internal droplets. This results from the non-trivial coupling between fluid velocity and internal cores (see Supplementary Fig. 3 for the structure of the flow field): continuous changes of droplet positions lead to significant variations of the local velocity field which, in turn, further modifies the motion of the cores in a typical self-consistent fluid–structure interaction loop.

**Suppression of crossings**. Before concluding, we observe that a viable route to prevent core crossings by keeping $N$ fixed, can be achieved by reducing the size (i.e. the diameter) of the inner drops, thus diminishing the area fraction $A_c$ they occupy within the emulsion.

In Supplementary Movie 11 we show, for example, the dynamics under Poiseuille flow of a four-core emulsion in which the inner drops, each of radius $R_i = 12$ lattice sites, are initially located in the lower half of the outer droplet of radius $R_O = 56$ lattice sites. This sets an area fraction $A_c \sim 0.2$. Following the scheme proposed in Fig. 3, one would expect the formation of a long-lived steady-state without core crossings. Indeed, once the Poiseuille flow is imposed, the cores are soon captured by the fluid vortex in the lower region where they remain confined and move with a persistent periodic motion. Such state, indicated as $\langle 0|1, 2, 3, 4\rangle$ is overall akin to $\langle 1, 2, 3|0\rangle$ shown in Fig. 3f, where three larger cores, with $A_c \sim 0.27$, are set into periodic motion by the fluid recirculation in the upper region of the emulsion.

If, unlike the previous case, cores of radius $R_i = 12$ lattice sites are placed symmetrically as in Fig. 2d, at late times the inner drop initially at the bottom (i.e. 4) is captured by the vortex in the lower half of the emulsion while those in the middle and at the top (i.e. 1, 2, 3) remain confined in the upper part (see Supplementary Movie 12). Clearly, this dynamics occurs without crossings. The emulsion finally attains a long-lived steady-state of the form $\langle 1, 2, 3|4\rangle$, analogous to the state $\langle 3|1, 2\rangle$ of Fig. 3e, i.e. a configuration in which an isolated core remains locked in a sector

of the emulsion and the remaining drops move periodically in the other one.

Hence the scheme proposed in Fig. 3 describes rather well the formation of these further steady states too, since they are long-lived, crossing-free and observed for $A_c < 0.35$.

## Discussion

Summarising, we have investigated the physics of a multi-core emulsion within a pressure-driven flow for values of Re and Ca typical of microfluidic experiments.

We have shown that, as long as $A_c$ and the $N$ are kept sufficiently low, the cores exhibit a periodic steady-state dynamics confined within a sector of the emulsion, reminiscent of a dancing couple, in which each dancer chases the partner. Our results strongly suggest that this peculiar behaviour is triggered and sustained by the internal vorticity which forms within the external droplet, whose interface acts as an effective bag confining the cores. The internal vorticity, in turn, is sustained by the heterogeneity of the micro-confined carrier flow, under the effect of the pressure drive.

As the area fraction and the number of cores increase, a more complex multi-body dynamics emerges. Due to unavoidable collisions between droplets, as combined with self-consistent hydrodynamic interactions, cores may leave the confining vortex and switch to the other one. Whenever this occurs, they either restore the planetary-like dynamics or temporarily aggregate with other cores to form unstable multi-droplets chains, which are repeatedly destroyed and re-established, due to the non-trivial coupling between the flow field and local variations of the phase field.

The jumps from one vortex to another are interpreted as entropic events, expressing the tendency of the system to maximise its entropy (propensity to motion) by filling voids, as they dynamically arise in this complex multi-body fluid–structure interaction.

Drawing inspiration from the occupation number formalism in statistical mechanics, we propose a classification of the NES that provides a transparent interpretation of their intricate dynamic behaviour. By denoting each dynamical state as $\langle \alpha_1, \ldots, \alpha_j|\alpha_{j+1}, \ldots, \alpha_N\rangle$, in which $j$ and $N-j$ distinct cores occupy the two sectors of the emulsion, respectively, this occupation-number formalism provides an elegant and transparent tool to (i) classify the various NES of the system and (ii) to describe the dynamic transitions among them.

On an experimental side, the aforementioned results can be realized by using standard microfluidic techniques, such as a glass capillary device with two distinct inner channels to form drops of different species within the emulsion[10,64]. Our system, in particular, could be mapped onto an emulsion in which cores, of diameter $D_i \sim 30\,\mu m$, are immersed in a drop of

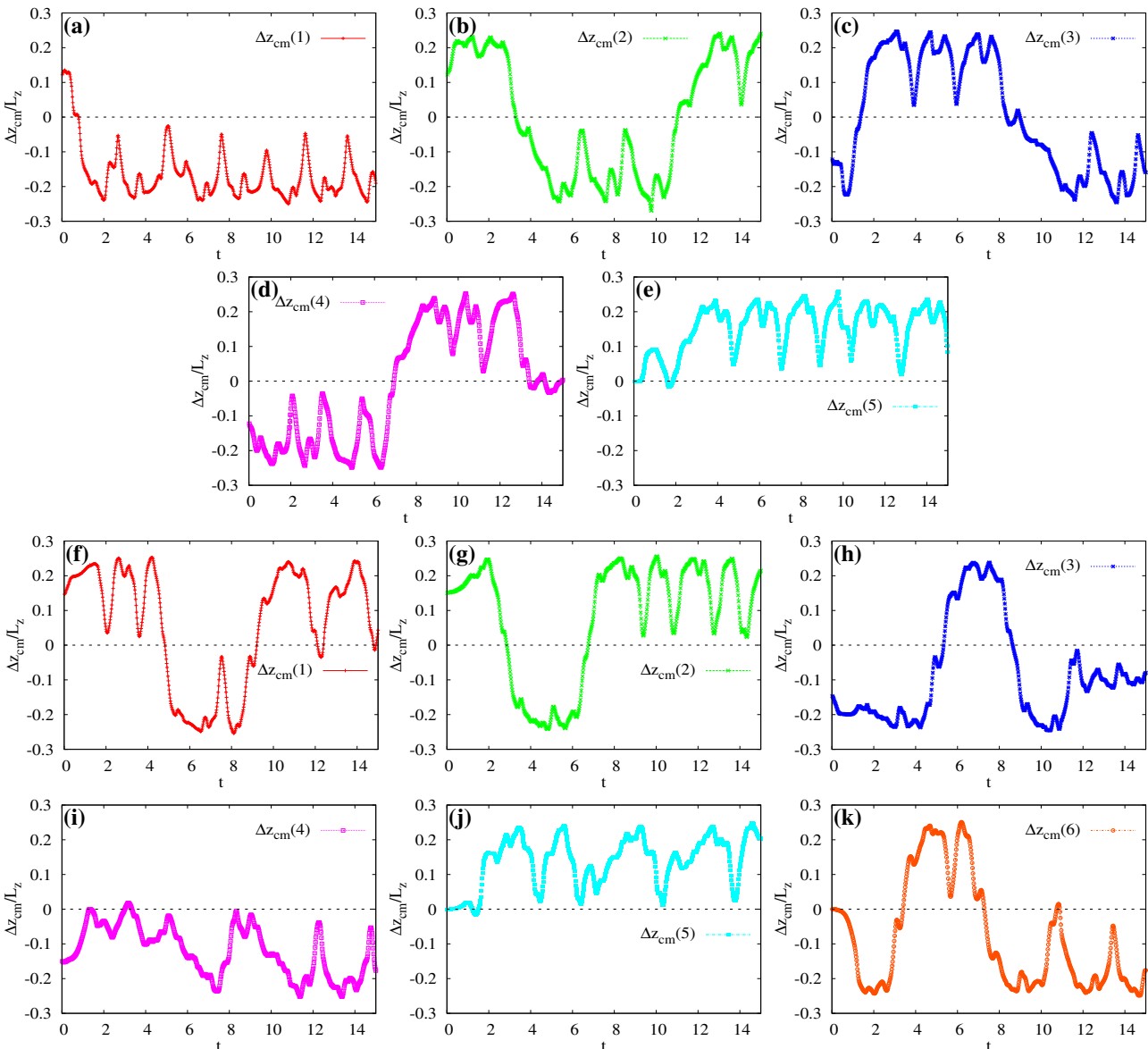

**Fig. 10 Multiple crossings in five and six-core emulsions. a–e** Time evolution of the displacement $\Delta z_{cm}$ of the cores in a five-core emulsion. Regular periodic motion survives for short periods of time and is temporarily broken by crossings of the cores towards either the top or the bottom of the emulsion. **f–k** Time evolution of $\Delta z_{cm}$ in a six-core emulsion. Although some cores travel along approximately similar trajectories (such as 1, red/plusses, and 2, green/crosses), there is no evidence of a persistent coupled periodic motion involving two (or more) drops. The black dotted line is a guide for the eye indicating the separation between the two regions of the emulsion.

diameter $D_O \sim 100\ \mu m$, with surface tension equal or higher than $\sigma \sim 1$ mN/m. Provided that the flow is kept within the laminar regime (to prevent turbulent-like behaviors), and the channel is sufficiently long (say order of millimiters) and large to minimize the squeezing of the emulsion, the steady-states discussed in Fig. 3 as well as their dynamics should be observed.

The insights on the non-equilibrium states delivered by the present study may prove useful to gain a deeper understanding of the role played by hydrodynamic interactions in multiple emulsions employed, for example, in fields like biology, pharmaceutics, and material science.

Recent microfluidic experiments, for instance, have been capable of encapsulating cells (the cores in our model) within aqueous droplets in the presence of other hosts, such as bacteria to study their pathogenicity[65]. Pinpointing the effect of the flow in these systems could shed light on the non-trivial dynamics governing the interaction among such biological objects, and

could potentially suggest how to tune the flow rate to control parameters of experimental interest, like reciprocal distance and position. Fluid–stucture interactions are also relevant in multiple emulsions used as carriers for drug delivery, since the release of the drug, generally stored within the cores, can be significantly influenced by the shape of the emulsion as well as by the structure of the fluid velocity within the shell[31]. The flow could facilitate, for example, the migration of a core towards regions exhibiting higher shape deformations, where a faster release is expected to occur[31]. In the context of material science, multiple emulsions are employed as building blocks to design droplet-based soft materials (such as tissues) with improved mechanical properties[1]. Monitoring the fluid flow is crucial here to achieve a uniform and regular arrangement of the droplets (generally required for these materials[34,35]) and to prevent structural modifications jeopardizing the design, an event occurring, for example, in the presence of multiple crossings. Our results support the view that

this latter effect can be significantly mitigate by keeping the area fraction of the cores sufficiently low. It would be also of interest to investigate how the dynamics in this regime is influenced by a change of the viscoelastic properties of the emulsion, achieved, for example, either by increasing the viscosity of the middle fluid (to harden the emulsion) or by partially covering the interface of the cores with a surfactant. This would be the case of Janus particles, which, owing to the takeover phenomena discussed in the text, would be periodically exposed to both front1–rear2 and front2–rear1 contacts.

Finally, one could also wonder whether periodic orbits of the cores as well as their transition may represent a potential mesoscopic-scale analogy with level crossing of atoms in quantum systems, along the lines pioneered by previous authors for the case of bouncing droplets on vibrating baths undergoing a tunnelling effect or orbiting with quantised diameters[66,67]. If so, one may even hope that multi-core emulsions may also provide hydrodynamic analogues of quantum materials. However at this stage, this is only a speculation which calls for a much more detailed and quantitative analysis.

## Methods
Following the approach of refs. [39,46], we describe the physics of a multi-core emulsion in terms of (i) a set of scalar phase field variables $\phi_i(\mathbf{r}, t)$, $i = 1, \ldots, N_d$ (where $N_d$ is the total number of droplets) accounting for the density of each droplet, and (ii) the global fluid velocity $\mathbf{v}(\mathbf{r}, t)$. The equilibrium properties are captured by a coarse-grained free-energy density

$$f = \frac{a}{4} \sum_i^{N_d} \phi_i^2 (\phi_i - \phi_0)^2 + \frac{k}{2} \sum_i^{N_d} (\nabla \phi_i)^2 + \epsilon \sum_{i,j,i<j} \phi_i \phi_j, \quad (1)$$

in which the first term guarantees the existence of two coexisting minima, $\phi_i = \phi_0$ inside the $i$th droplet and $\phi_i = 0$ outside, while the second one determines the interfacial tension. The two positive constants $a$ and $k$ set the surface tension $\sigma = \sqrt{8ak/9}$ and the interface thickness $\xi = 2\sqrt{2k/a}$ of the droplets. Finally, the last contribution is a soft-core repulsive term whose magnitude is controlled by the (positive) constant $\epsilon$. Such term mimics the presence of a surfactant adsorbed onto the droplet interfaces and, if $\epsilon$ is sufficiently high (tyically higher than 0.005), it prevents droplets merging.

The dynamics of the order parameters $\phi_i$ is controlled by a set of Cahn–Hilliard equations

$$\frac{\partial \phi_i}{\partial t} + \nabla \cdot (\phi_i \mathbf{v}) = M \nabla^2 \mu_i, \quad (2)$$

where $M$ is the mobility and $\mu_i = \partial f / \partial \phi_i - \partial_\alpha \partial f / \partial(\partial_\alpha \phi_i)$ is the chemical potential (Greek letters denote Cartesian components).

The fluid velocity $\mathbf{v}$ is governed by the Navier–Stokes equation which, in the incompressible limit, reads

$$\rho \left( \frac{\partial}{\partial t} + \mathbf{v} \cdot \nabla \right) \mathbf{v} = -\nabla p + \eta \nabla^2 \mathbf{v} - \sum_i \phi_i \nabla \mu_i. \quad (3)$$

In Eq. (3), $\rho$ is the density of the fluid, $p$ is the isotropic pressure, and $\eta$ is the dynamic viscosity. Equations (2) and (3) are numerically solved by using a hybrid lattice Boltzmann (LB) approach[68,69], in which a finite difference scheme, adopted to integrate Eq. (2), is coupled to a standard LB method employed for Eq. (3). Further details about numerical implementation and thermodynamic parameters can be found in Supplementary Note 1.

We finally provide an approximate mapping between our simulation parameters and real physical values. Simulations are run on a rectangular mesh of size varying from $L_y = 600/800$ (length of the channel) to $L_z = 100/170$ (height of the channel), in which droplets, of radius ranging from 15 to 56 lattice sites, are included. Lattice spacing and time-step are $\Delta x = 1$ and $\Delta t = 1$. These values would correspond to a microfluidic channel of length ~1 mm, in which droplets, with diameter ranging between 30 and 100 μm and surface tension $\sigma \sim 1$ mN/m, are set in a fluid of viscosiy $\simeq 10^{-1}$ Pa s. By fixing the length scale, the time scale, and the force scale as $L = 1$ μm, $T = 10$ μs, and $F = 10$ nN, a velocity of $10^{-2}$ in simulation units corresponds approximately to a droplet speed of 1 mm/s. The Reynolds number (defined as $Re = \frac{\rho D_O v_{max}}{\eta}$, where $D_O$ is the diameter of the shell) ranges approximately between 1 ($\Delta p = 4 \times 10^{-4}$ and $v_{max} \simeq 0.01$) and 5 ($\Delta p = 10^{-3}$ and $v_{max} \simeq 0.025$), while the capillary number (defined as $Ca = \frac{v_{max} \eta}{\sigma}$) ranges between 0.1 and 1. These values ensure that inertial effects are mild and are in good agreement with those reported in previous experiments[1] and simulations[52].

## Data availability
All data are available upon request from the authors.

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

## Acknowledgements

A.T., A.M., M.L., F.B. and S.S. acknowledge funding from the European Research Council under the European Union's Horizon 2020 Framework Programme (No. FP/2014–2020) ERC Grant Agreement No.739964 (COPMAT).

## Author contributions

A.T., A.M., M.L., F.B, S.S, S.A., M.M., and D.A.W. conceived the research. A.T. and S.S. designed the project. A.T. run simulations and processed data, and with A.M. and S.S. analyzed the results. A.T. wrote the paper with contributions from A.M., M.L, F.B., S.S., S.A., M.M., and D.A.W.

## Competing interests

The authors declare no competing interests.
