## [Peer Review File · Nature Communications]

REVIEWER COMMENTS

Reviewer #1 (Remarks to the Author):

This manuscript investigated the physics of multi-core emulsions flowing in microfluidic channels and reported the numerical evidence of a surprisingly rich variety of new driven non-equilibrium states. The work is very interesting and has been well done. It could be accepted after some points are addressed more clearly:

- 1) The initial locations of the cores are very important to affect rheological behaviors of the multi-core emulsions moving in a straight micro-channel. Although the authors have discussed a little about this issue, it is still not enough since initial locations of cores are extremely diverse relatively to the principal direction of the flow.
- 2) When the numbers of cores are even, the systems might be symmetric, for instance, fig 2b and 2d. For these symmetric systems, there is a transition from the symmetry to the asymmetry during the moving in the channel (see fig 4). The physical mechanism of the symmetry-to-asymmetry transition should be addressed more clearly. The authors should give an explanation to the instability of the physical phenomenon and of the numerical calculation.
- 3) In experiments, when two droplets collide with each other, they might merge. In this work, the numerical calculation seems not capture the coalescence of two colliding cores, for instance fig 4. The authors should give an explanation of this issue.
- 4) Is it possible that the results shown in this work could be testified by some experiments?

Reviewer #2 (Remarks to the Author):

The article describes numerical simulations of the motion of multi-core emulsions (i.e. droplets embedded within a larger drop) into a microfluidic channel. Numerical simulations with 1 to 6 cores embedded into a single drop are presented. It is found that the nature of the motion of the core(s) evolves as the number of cores is increased: it is stationary with 1 core, stationary or periodic with two cores, close to periodic with 3 cores, and pre-chaotic or chaotic with a higher number of cores. The paper is well written and I have only minor comments on the form and content.

I did find the results interesting, but my feeling is that it would appeal to a rather limited audience. It is interesting to study the evolution of the motion toward chaotic regimes, but the implications of the results is not clear to me. Perhaps the authors should give a bit more details on the implications of their findings? It is proposed for example that multi core emulsions may provide analogues of quantum materials, but this is quite speculative at this stage.

More specific comments or questions :

* It is argued that the transition between short-lived non-equilibrium states is due to the high value of A_c . Is it really A_c that matters, or N ? Or both in different ways? I understand that for example the $\langle 1,2,3,4|0 \rangle$ state is indeed not realizable if A_c is too large, but the unsteady dynamics may also be related to N rather than A_c . Has this been tested (i.e. by changing the diameter of the cores).

* Several parameters have not been explicitly defined in the text.

- the velocity and sizes are given in non-dimensional forms, but the way the problem has been non-dimensionalized is not always clear. For example, the external droplet and cores radii are all larger than 1. What lengthscale has been used to make them non-dimensional? Please clarify.

- L_x and L_z are not defined in the text. We can guess that these are measures of the size of the drop along the x and z directions, but how are they defined exactly?

- on figure 1, it is not clear how the vorticity is estimated. Is it some average of the magnitude of the vorticity within the drop? Please explain how it has been estimated. The text in the caption of figure 1 suggests that what is shown is an estimate of the vorticity given by v_{\max}/L_z . If that the case, the plot of

omega as a function of Re can probably be removed. If L_z does not vary much in the explored range of Re, then it is straightforward that v_{\max}/L_z is proportional to Re. Also, perhaps a better estimate of the vorticity could be given by $(v_{\max} - v_x(-L_z/2))/(L_z/2)$ (where $v_x(-L_z/2)$ is the velocity at $z=-L_z/2$).

* On figure 8, the bottom figure should perhaps be put with figure 7, or as a separate figure (it shows the $\langle 3|1,2\rangle$ state and not the $\langle 1,2,3|0\rangle$ state described on the top figure).

Also, the top figure is not referred to in the text (it should be referred to on the first paragraph of the right column of page 6).

* I have highlighted a few typos on the pdf (see file attached).

REVIEWERS COMMENTS

Reply to Reviewer #1

Comment: *This manuscript investigated the physics of multi-core emulsions flowing in microfluidic channels and reported the numerical evidence of a surprisingly rich variety of new driven non-equilibrium states. The work is very interesting and has been well done.*

Our response: We warmly thank the Reviewer for her/his appreciative remarks on our work.

Comment: *1) The initial locations of the cores are very important to affect rheological behaviors of the multi-core emulsions moving in a straight micro-channel. Although the authors have discussed a little about this issue, it is still not enough since initial locations of cores are extremely diverse relatively to the principal direction of the flow.*

Our response: We thank the Reviewer for raising this valuable point.

As long as the number of cores remains low, such as in Fig.2 a-b-c, the steady states attained by the emulsion under flow are essentially those reported in Fig.3 a-b-c-d-e-f. As correctly noted by the Reviewer, the existence of such states is sensitive to the initial location of the cores within the emulsion. For example, unlike the state $\langle 1, 2|0 \rangle$, the state $\langle 1|2 \rangle$ has been obtained by confining each core in a different sector of the emulsion. Also, the state $\langle 1, 2, 3|0 \rangle$ can be achieved by confining the three cores in the upper part of the emulsion. These aspects were shortly mentioned in the previous version, and they have been discussed more extensively in the revised one.

As the number of cores is increased, the physics gets more complex and a pre-chaotic behaviour is seen to emerge. For these cases, however, the initial location of the cores seems to play a minor role in the drive towards the steady state. Considering, for instance, a four-core emulsion (Fig.1d), the steady states are essentially those reported in Fig.3 g-h, regardless of the initial position of the cores. Indeed, a further simulation (see movie M8.avi) shows that the steady state $\langle 1, 3|2, 4 \rangle$ occurs even if, as an example, the four cores start in the lower part of the emulsion.

Yet, different steady states can be observed if we change the *size* (i.e. the diameter) of the cores, rather than their initial position. In a four-core emulsion, for example, we observe that reducing the diameter of the inner drops

with $A_c \sim 0.2$, yields to the formation of novel long-lived nonequilibrium states, such as $\langle 0|1, 2, 3, 4\rangle$, without any crossings. However, such states share a similar dynamic behavior of those reported in Fig.3 at area fraction lower than ~ 0.35 (like the state $\langle 1, 2, 3|0\rangle$), and their formation is well described by the scheme proposed therein.

We kindly refer the Reviewer to the response to the second Referee, where this point is discussed more extensively.

Comment:2) *When the numbers of cores are even, the systems might be symmetric, for instance, fig 2b and 2d. For these symmetric systems, there is a transition from the symmetry to the asymmetry during the moving in the channel (see fig 4) . The physical mechanism of the symmetry-to-asymmetry transition should be addressed more clearly. The authors should give an explanation to the instability of the physical phenomenon and of the numerical calculation.*

Our response: The transition from a symmetric (equilibrium) state towards an asymmetric (nonequilibrium) one, is essentially governed by the coupling between fluid velocity \mathbf{v} and gradient of the phase field $\nabla\phi$.

Considering, for example, the case shown in Fig.4, the two cores, initially equilibrated like in Fig.2b, are carried rightwards once the Poiseuille flow is imposed. When the external droplet attains a bullet-like shape at the steady state, the cores accumulate at the leading edge of the outer interface and arrange in a row-like one dimensional configuration. Such state is however unstable to weak perturbations of the velocity field, which exhibits local deviations from the unidirectional pattern (observed, for example, at the leading edge of a core-free emulsion under flow, see Fig.1c), in particular within and in the surroundings of the cores.

In other words, the interfaces locally modify the uniform structure of the velocity field which, in turn, triggers the motion of a drop either upwards or downwards. Once this occurs, the core is captured by the fluid vortex and is set into motion. An analogous mechanism holds for the other inner drop.

We thank the Reviewer for the valuable comment about this point. The physics behind the symmetry-to-asymmetry transition has now been clarified in the revised text.

Comment:3) *In experiments, when two droplets collide with each other, they might merge. In this work, the numerical calculation seems not capture the*

coalescence of two colliding cores, for instance fig 4. The authors should give an explanation of this issue.

Our response: As correctly pinpointed by the Reviewer, droplet merging can occur in experiments of multiple emulsions, especially in the presence of hard collisions under flow. However, in many practical situations, the aim is to study how the complex hydrodynamic interactions among distinct droplets affect the mechanics of materials composed, for example, by a regular array of emulsions with predefined morphology (see, for example, L. L. A. Adams et al., *Soft Matter* **8**, 10719 (2012)). In these cases it is crucial to prevent droplet coalescence.

Merging can be essentially inhibited (or considerably reduced) by using surfactant solutions adsorbed onto the droplet interface (see, for example, Wang et al., *Lab on Chip* **11**, 1587-1592, (2011); A. R. Abate and D. A. Weitz, *Small* **5**, 2030-2032 (2019); A. S. Utada et al., *Science* **308**, 537-541 (2005); T. Sheth, et al. *Nature Reviews Materials* **5**, 214-228 (2020)). These are precisely the systems we are considering.

In our framework, the suppression of droplet merging is thermodynamically controlled by the term $\epsilon \sum_{i,j,i < j} \phi_i \phi_j$ appearing in the free energy (Eq.1 in Methods). As long as ϵ is high enough, this soft-core repulsive contribution prevents droplet coalescence and stabilizes the emulsion. In our simulations, we have set $\epsilon = 0.05$ (see Supplemental Material), and we have empirically observed that merging is suppressed whenever $\epsilon \geq 0.005$. In the limit $\epsilon \rightarrow 0$ droplet coalescence and Ostwald ripening occur.

Finally, note that such description holds because we are within the laminar regime, in which hard collisions among cores are fairly unlikely.

We have clarified these aspects in Introduction and Methods in the revised version of the manuscript.

Comment:4) *Is it possible that the results shown in this work could be testified by some experiments?*

Our response: Yes, it is certainly possible by using rather standard microfluidic techniques for the production of multiple emulsions. We actually planned, in the Weitz lab, a series of dedicated experiments, which, due to the pandemic, have been inevitably postponed.

A monodisperse multi-core emulsion can be produced, for example, by using a glass capillary device with two distinct inner channels to form drops of different species within the emulsion. Their number can be controlled by

tuning the flow rate, which has to be determined for each set of fluid (see, for example, L.L.A. Adams et al., *Soft Matter* **8**, 10719 (2012)). If, for example, one decreases the flow rate of the outermost phase while keeping the other two flow rates equal, one will get large mother droplets, that contain an increasingly large number of inner cores. Such technique allows for the separate injection of different oil phases which are encapsulated within an aqueous phase (the middle fluid) and finally emulsified in the outer fluid. The resulting emulsion is then collected in the exit channel and dragged by a pressure-driven flow.

Based on our results, we expect that, in order to observe the steady states described in Fig.3 (core rotations, crossings, chaotic behaviour), at least three conditions must be met: i) the diameter of the cores should remain approximately $1/3$ (or lower) than the one of the outer droplet (to avoid crowding), ii) the flow has to be within the laminar regime (to prevent turbulent-like behaviors) and iii) the size of the channel should be long enough (say order of millimeters) and sufficiently large to minimize the squeezing of the emulsion. In addition, the range of values of experimental parameters, such as flow rates, droplet surface tension, viscosity of the fluids (to name but a few), could affect the results. A mapping with values used in simulations is also provided in the section Methods.

We finally mention a couple of potential sources of mismatch between simulations and experiments. The first one pertains to the design of the device. Droplets fabricated in PDMS devices are usually strongly confined along the direction of the channel, looking more like pancakes than actual spheres. This is often a desired condition, since the flat surfaces of the pancake-shaped droplet makes it easier to image it. From a pure fluid-mechanics perspective, this constraint could be released, but a purposeful design is necessarily required. The second one might come from the 3d nature of experiment, where an out-of-plane motion of the cores might occur, especially when many-body interactions are relevant. However, if the flow is kept within the laminar regime, this issue is comparatively mild and our simplified 2d description is expected to capture the essential features of the physics point.

A prospective experimental realisation of our system is discussed in the conclusion of the revised version.

REVIEWERS COMMENTS

Reply to Reviewer #2

Comment: *The article describes numerical simulations of the motion of multi-core emulsions (i.e. droplets embedded within a larger drop) into a microfluidic channel. Numerical simulations with 1 to 6 cores embedded into a single drop are presented. It is found that the nature of the motion of the core(s) evolves as the number of cores is increased: it is stationary with 1 core, stationary or periodic with two cores, close to periodic with 3 cores, and pre-chaotic or chaotic with a higher number of cores. The paper is well written and I have only minor comments on the form and content.*

Our response: We thank very much the Reviewer for her/his positive assessment of our work.

Comment: *I did find the results interesting, but my feeling is that it would appeal to a rather limited audience. It is interesting to study the evolution of the motion toward chaotic regimes, but the implications of the results is not clear to me. Perhaps the authors should give a bit more details on the implications of their findings?*

Our response: In the revised version, we have extensively discussed the potential implications of our results in other sectors of science, besides physics. Over the last few years, the research in the manufacturing of multiple emulsions has been boosted by dramatic advances in droplet-based microfluidics experiments. Such experiments have provided a wide number of innovative platforms to develop complex design of these systems under highly controlled conditions. On the other hand, fundamental studies on the dynamics of multiple emulsions are more recent, and have raised considerable interest mainly in soft material science, biology and pharmaceuticals. Thus, besides soft matter and the physics of fluids, our work essentially also embraces the three aforementioned broad fields of science. Let us discuss the point in some more detail.

Soft materials: Multiple emulsions can generally be used in material science as building blocks to produce droplet-based soft tissues or foamy-like manufactures. This is because, unlike their core-free counterpart, the mechanical properties of their inner structure can be modulated in a more controlled manner (see, for example, Utada et al., Science **308**, (2005) and L.L.A.

Adams et al., *Soft Matter* **8**, 10719 (2012)). Since the structural integrity and mechanical stability of these composite materials crucially depends on shape, size and response to deformations of the emulsions, the investigation of rheological properties, as well as fluid-structure interactions is fundamental to guarantee a correct design of these materials.

In this context, the transition towards a pre-chaotic regime (occurring through crossings) would point to a rather sharp modification of the physical properties of a predetermined material, an effect that one may wish to avoid in experiments, especially if a uniform and regular design is needed. Besides keeping the core area fraction sufficiently low (as suggested by our results), suppression of crossings could be likely achieved by either increasing the viscosity of the middle fluid (to harden the emulsion) or by modifying the viscoelastic properties of the interfaces of the cores (by using, for example, different surfactants covering the droplet with a solid-like “skin”). The latter scenario, in particular, could be modelled by endowing each phase field ϕ_i with its own repulsive strength ϵ_i , to mimic the simultaneous presence of different surfactants. We are currently investigating both strategies, and they will be the subject of future works.

Biology: In this sector, a number of experiments, particularly in the Weitz lab, aims at studying the interaction among biological objects, such as cells, bacteria and antibiotics, confined within aqueous droplets.

For example, cells of a human tissue (to be modelled as cores in our description) can be co-encapsulated with unknown bacteria within a droplet, in order to assess the degree of pathogenicity of the latter. An analogous experimental setup is used to produce organoids, a simplified version of an organ in vitro, obtained by initially encapsulating cells within aqueous droplets and then sorting the droplets with the desired number and composition of starting cells. From a physical standpoint, understanding how hydrodynamics mediate the interactions among such objects could shed light on their complex coupling with the flow and potentially suggest suitable strategies to tune the flow rate in such a way as to control reciprocal distance and motion.

Pharmaceutics: It is also well known that bioencapsulation techniques have provided a very promising strategy for the development of therapeutic treatments for diabetes, renal failure and cancer (see G. Orive et al., *Nature Medicine* **9**, 104 (2003)). This inevitably calls into play the use of multiple emulsions as efficient carriers for drug delivery. Indeed, these systems possess several key features, such as an enhanced control on the release of the sub-

stances due to the middle fluid, a highly compartmental structure for drug storing, as well as a higher resistance to mechanical failure with respect to the core-free counterpart.

Thus, the investigation of the dynamics of multi-core emulsions in microchannels is helpful to shed light on crucial processes in pharmaceuticals, such as the controlled release of injected drugs. This effect is known to be significantly affected by the shape of the emulsion, as well as by the structure of the fluid velocity within the shell (see G. Pontrelli et al., Phys. Rev. E **102**, 023114 (2020)). The flow could facilitate, for example, the migration of the carrier towards regions of the emulsion exhibiting higher shape deformations (or higher interface curvature), where a faster release is expected to occur. Finally, our work is also expected to raise the general interest of soft matter scientists and of the fluid-dynamic community, since it has uncovered, in an everyday life example of soft fluid, a series of novel dynamical features which, besides holding a theoretical relevance on their own, are potentially realisable in future microfluidic experiments.

All these considerations have been discussed in the revised version.

Comment: *It is proposed for example that multi core emulsions may provide analogues of quantum materials, but this is quite speculative at this stage.*

Our response: Hydrodynamic quantum analogues were first introduced in the seminal papers of Couder's group, such as Couder et al. Nature **437**, 208 (2005), Eddi et al., Phys. Rev. Lett. **102**, 240401 (2009) and Eddi et al., Phys. Rev. Lett. **108** 264503 (2012), in which fluid droplets have been found to exhibit dynamic features analogous to quantum mechanical systems, and most notably the famous DeBroglie pilot-wave formulation of quantum mechanics.

For example, in PRL (2009) it was shown that a fluid droplet bouncing on a vibrated bath may undergo a tunnelling effect, analogous to quantum tunnelling, when it collides with barriers, with a crossing probability decreasing with increasing barrier width. In PRL (2012) it was further shown that two walking fluid droplets on a vibrating fluid surface orbit around each other in a stable configuration with quantised orbit diameters.

Inspired by such results, we argued that periodic orbits of the cores as well as their transition from upper to lower region (or viceversa) of the emulsion might represent a potential mesoscopic-scale analogy with level crossing of atoms in quantum systems. On a speculative vein, one may also wonder

whether the vortex-structure of the flow within the capsule might serve as a non-linear pilot wave associated with the Gross-Pitaevski equation describing quantum fluids and most notably Bose-Einstein condensates. However, we fully agree with the Reviewer that the connection between multi-core emulsion and quantum materials remains highly speculative at the present stage, and calls for a much deeper and quantitative investigation. As a result, we have shortened the discussion on this point and expanded instead the discussion on links to the aforementioned research on soft materials and biotech applications.

Comment: * *It is argued that the transition between short-lived non-equilibrium states is due to the high value of A_c . Is it really A_c that matters, or N ? Or both in different ways ? I understand that for example the $\langle 1, 2, 3, 4|0 \rangle$ state is indeed not realizable if A_c is too large, but the unsteady dynamics may also be related to N rather than A_c . Has this been tested (i.e. by changing the diameter of the cores).*

Our response: We thank very much the Reviewer for raising this valuable point.

In order to assess more carefully the role played by N , as well as its interplay with A_c , we have run a selected number of new simulations, in particular when the number of cores is sufficiently high (i.e. $N \geq 4$). Indeed, as long as both N and A_c remain *low* (such as in Fig.2 a-b-c), the steady states attained by the emulsion under flow are those reported in Fig.3 a-b-c-d-e-f. See also, for example, movie R2.avi, in which an emulsion with two cores of radius $R = 8$ lattice sites (smaller than the one considered in the paper) is subject to a Poiseuille flow. A steady-state of the form $\langle 1, 2|0 \rangle$ appears at a later time.

As correctly pinpointed by the Reviewer, a more complex scenario emerges when N and A_c increase. Cases discussed in Fig.3 g-h-i-j-k-l essentially pertain to a regime with *high* values of A_c (> 0.35) and *high* values of N (≥ 4).

The typical dynamics of a regime with *low* A_c (lower than 0.35) but *high* N (higher than 3) is shown in movie M11.avi. Here a four-core emulsion, in which each core has radius $R = 12$ lattice sites (smaller than the values considered in previous runs, with $R = 17$, see supporting material), is subject to a Poiseuille flow. Cores are located in the lower part of the outer droplet, with radius $R = 56$ lattice sites, corresponding to an area fraction $A_c \sim 0.2 < 0.35$. Once the Poiseuille flow is applied, the cores remain confined

within that region (no crossings occur) and rotate coherently, leading to a new nonequilibrium state indicated as $\langle 0|1, 2, 3, 4 \rangle$.

If, unlike the previous case, the cores are initially located symmetrically as in Fig.2d, the steady-state $\langle 1, 2, 3|4 \rangle$ is found at late time and, once again, no core crossings are observed (see movie M12.avi).

Such results are however in agreement with the scheme proposed in Fig.3, since these states are both long-lived ones and found at area fraction A_c below 0.35. Indeed, the new states are overall akin to the ones obtained for $N = 3$; in particular, the state $\langle 0|1, 2, 3, 4 \rangle$ shares a similar dynamic behavior with state $\langle 1, 2, 3|0 \rangle$, as well as the state $\langle 1, 2, 3|4 \rangle$ with $\langle 3|1, 2 \rangle$ (i.e. an isolated drop in one sector of the emulsion and the remaining ones confined in the other sector).

Since in both cases core crossings are suppressed, we have devoted a full new section of the manuscript to describe this effect.

To summarize, we now distinguish two regimes. The first one accounts for states at low area fraction A_c (lower than 0.35) and low number of cores N (< 4), and a second one which includes states at high A_c (higher than 0.35) and high N (≥ 4). The first group encompasses the states reported in Fig.3 a-f, while the second one comprises those in Fig.3 g-l.

Finally, we note that we have not considered cases at very high A_c , such as emulsions with a high number of cores with $D_i \ll D_O$ (diameter of inner cores and outer droplet) at the close-packing fraction limit or, alternatively, systems with a few cores where $D_i \sim D_O$ (such as two large inner drops). The detailed investigation of the physics of such multi-core emulsion calls for a separate work on its own and will make the subject of future studies.

Comment: * *Several parameters have not been explicitly defined in the text. - the velocity and sizes are given in non-dimensional forms, but the way the problem has been non-dimensionalized is not always clear. For example, the external droplet and cores radii are all larger than 1. What lengthscale has been used to make them non-dimensional ? Please clarify.*

Our response: All quantities in the paper are given in simulation units, while the mapping to physical values has been reported in the section Methods, the last of the main text. The details of the simulation parameters are described in the supporting material.

We have basically considered droplets of radius ranging from 15 to 56 lattice units, placed within a rectangular mesh of size varying from $L_y = 600 \div 800$

(longitudinal dimension) and $L_z = 100 \div 170$ (transversal dimension). Lattice spacing and time-step are $\Delta x = 1$ and $\Delta t = 1$. Thus, for instance, the droplet in Fig.1 has radius $R = 30\Delta x = 30 \times 1 = 30$ lattice sites. The mapping to physical units is built by fixing the typical length scale, time scale and force scale to $L = 1\mu\text{m}$, $T = 10\mu\text{s}$ and $F = 10\text{nN}$. In particular, $\Delta x = L$ and $\Delta t = T$, hence a droplet of radius $R = 30$ in simulation units corresponds to a radius $\sim 30\mu\text{m}$ in physical units. Also, a speed around 10^{-3} in simulation units would correspond to $10^{-3}\Delta x/\Delta t \sim 0.1\text{mm/s}$ in physical units. We have further clarified these aspects in the section Methods of the revised manuscript.

Comment:- L_x and L_z are not defined in the text. We can guess that these are measures of the size of the drop along the x and z directions, but how are they defined exactly ?

Our response: We apologise for the misunderstanding. L_y and L_z are actually the longitudinal and transversal dimensions of our microfluidic channel, not of the droplets (their values are reported in the SI material). This is now clarified in the Methods section of the revised paper.

Comment:- on figure 1, it is not clear how the vorticity is estimated. Is it some average of the magnitude of the vorticity within the drop ? Please explain how it has been estimated. The text in the caption of figure 1 suggests that what is shown is an estimate of the vorticity given by v_{max}/L_z . If that the case, the plot of omega as a function of Re can probably be removed. If L_z does not vary much in the explored range of Re , then it is straightforward that v_{max}/L_z is proportional to Re . Also, perhaps a better estimate of the vorticity could be given by $(v_{max} - v_x(-L_z/2))/(L_z/2)$ (where $v_x(-L_z/2)$ is the velocity at $z = -L_z/2$).

Our response: Thank you for spotting this issue that was not clear enough in the manuscript. The approximate estimate of the magnitude of the vorticity stems from the idea that in our simulations the drop size is overall comparable with the thickness L_z of the channel and the velocity at the edge of the vortex is approximately equal to the maximum value v_{max} in the middle of the channel (see Fig.1c). Thus one may write $\omega \sim v_{max}/L_z$, which might be a way of assessing ω from parameters directly accessible and controllable in experiments.

However, we agree with the Reviewer that, if L_z is constant (like in this case),

the proportionality is straightforward. Hence we decided to follow the advice and, since no essential information is overall lost, we removed the plot in the revised version.

Comment: * *On figure 8, the bottom figure should perhaps be put with figure 7, or as a separate figure (it shows the $\langle 3|1,2 \rangle$ state and not the $\langle 1,2,3|0 \rangle$ state described on the top figure). Also, the top figure is not referred to in the text (it should be referred to on the first paragraph of the right column of page 6).*

Our response: Thanks for spotting this issue. The top part of Fig.8 (the speeds of each core) refers to the state $\langle 3|1,2 \rangle$ reported in Fig.7, as well as the bottom one. We have clarified this issue in the main text and properly referred to it in the revised version.

Comment: * *I have highlighted a few typos on the pdf (see file attached).*

Our response: Thank you for spotting them. We have corrected the typos in the revised version.

REVIEWERS' COMMENTS

Reviewer #1 (Remarks to the Author):

The revised version of your manuscript is fine and could be accepted by NC.

Reviewer #3 (Remarks to the Author):

I am satisfied by the answers to the comments and the changes made to the manuscript. I believe the manuscript can be accepted in its current form.

REVIEWERS COMMENTS

Reply to Reviewer #1 and #2

Comment: *Reviewer 1: The revised version of your manuscript is fine and could be accepted by NC.*

Reviewer 2: I am satisfied by the answers to the comments and the changes made to the manuscript. I believe the manuscript can be accepted in its current form.

Our response: We warmly thank both Reviewers for the positive comments and for supporting the publication of our work.